# FDA orphan drug designations for lysosomal storage disorders – a cross-sectional analysis

**Sven F. Garbade**[1], **Matthias Zielonka**[1], **Konstantin Mechler**[2], **Stefan Kölker**[1], **Georg F. Hoffmann**[1], **Christian Staufner**[1], **Eugen Mengel**[3], **Markus Ries**[1,4,5]*

1 Division of Pediatric Neurology and Metabolic Medicine, Center for Pediatric and Adolescent Medicine, University Hospital Heidelberg, Heidelberg, Germany, 2 Department of Child and Adolescent Psychiatry and Psychotherapy & Department of Addictive Behavior and Addiction Medicine, Medical Faculty Mannheim, Central Institute of Mental Health, University of Heidelberg, Mannheim, Germany, 3 SphinCS GmbH, Science for LSD, Hochheim, Germany, 4 Center for Rare Diseases, University Hospital Heidelberg, Heidelberg, Germany, 5 Center for Virtual Patients, Medical Faculty, University of Heidelberg, Heidelberg, Germany

* markus.ries@uni-heidelberg.de

**Data Availability Statement:** All data are in the paper and supporting information.

**Funding:** SphinCS GmbH provided support in the form of salary for EM, but did not have any

## Abstract

### Purpose

To provide a quantitative clinical-regulatory insight into the status of FDA orphan drug designations for compounds intended to treat lysosomal storage disorders (LSDs).

### Methods

Assessment of the drug pipeline through analysis of the FDA database for orphan drug designations with descriptive and comparative statistics.

### Results

Between 1983 and 2019, 124 orphan drug designations were granted by the FDA for compounds intended to treat 28 lysosomal storage diseases. Orphan drug designations focused on Gaucher disease (N = 16), Pompe disease (N = 16), Fabry disease (N = 10), MPS II (N = 10), MPS I (N = 9), and MPS IIIA (N = 9), and included enzyme replacement therapies, gene therapies, and small molecules, and others. Twenty-three orphan drugs were approved for the treatment of 11 LSDs. Gaucher disease (N = 6), cystinosis (N = 5), Pompe disease (N = 3), and Fabry disease (N = 2) had multiple approvals, CLN2, LAL-D, MPS I, II, IVA, VI, and VII one approval each. This is an increase of nine more approved drugs and four more treatable LSDs (CLN2, MPS VII, LAL-D, and MPS IVA) since 2013. Mean time between orphan drug designation and FDA approval was 89.7 SD 55.00 (range 8–203, N = 23) months.

### Conclusions

The drug development pipeline for LSDs is growing and evolving, with increased focus on diverse small-molecule targets and gene therapy. CLN2 was the first and only LSD with an approved therapy directly targeted to the brain. Newly approved products included "me-too"–enzymes and innovative compounds such as the first pharmacological chaperone for the treatment of Fabry disease.

additional role in the study design, data collection and analysis, decision to publish, or preparation of the manuscript. The specific roles of this author are articulated in the 'author contributions' section.

**Competing interests:** I have read the journal's policy and the authors of this manuscript have the following competing interests: SG, SK, and CS have no potential conflicts of interest to declare with respect to the research, authorship, and/or publication of this article. KM has served as investigator in clinical trials conducted by Emalex, Gedeon Richter, Lundbeck, Shire, Sunovion and Teva, plus in European Union funded projects. GFH received lecturing fees from Danone and Takeda. EM has received honoraria and/or consulting fees from Actelion, Alexion, BioMarin, Orphazyme, Sanofi Genzyme, and Shire. MR received consultancy fees or research grants from Alexion, GSK, Oxyrane and Shire. EM is managing partner of SphinCS GmbH, a private clinical research institute, which had research grants from Sanofi Genzyme, Takeda, Alexion, Orphazyme and Idorsia. This does not alter our adherence to PLOS ONE policies on sharing data and materials.

# Introduction

Lysosomal storage disorders (LSDs) are a group of more than 50 inherited, multisystemic, progressive conditions caused by a genetic defect that results in the progressive accumulation of complex non-metabolized substrates in the lysosomes of cells, tissues and organs, inducing distinct but heterogeneous somatic and neurological disease phenotypes [1–7]. In general, lysosomal storage disorders lead to significant morbidity and decreased life expectancy. Reported prevalences of LSDs in industrialized countries range between 7.6 per 100,000 live births (= 1 in 13,158) and 25 per 100,000 live births (= 1 in 4000) [8–11]. Some LSDs are treatable and the drug development in the field has traditionally been very active and dynamic after the successful development of enzyme replacement therapy in Gaucher disease which seeded further innovation [1, 12, 13]. The development of new compounds and new concepts of treatment for lysosomal storage disorders has been very dynamic. Therefore, the purpose of the present paper is to precisely analyze the most recent advances and novel trends in orphan drug development for lysosomal storage diseases as documented in the FDA Orphan Drug Product designation database.

# Methods

The FDA orphan drug database was accessed over the internet at the following address http://www.accessdata.fda.gov/scripts/opdlisting/oopd/. Search criteria were "all designations" from 1 January 1983 until 10 May 2019, i.e., all data entries until 10 May 2019 were taken into account (N = 4979). The output format was an excel file which was downloaded on a local computer. Orphan designations for lysosomal storage diseases were extracted with pertinent keywords (N = 124) [1]. STROBE criteria (S1 Checklist) were respected [14].

## Definitions

Pharmacological compounds were categorized based on their chemical structures into the following classes, listed in alphabetical order: "enzyme", "enzyme/small molecule combination", "gene therapy", "polymer", "protein (other than enzyme)", and "small molecule" [1]. A small molecule was defined as a compound with a molecular weight below 900 Da [15]. In addition, compounds were further grouped into functionally meaningful subtypes based on their biochemical properties, molecular mechanisms of action, or gene therapy platforms, i.e., (in alphabetical order): "AAV vector", "adjunctive therapy", "anaplerotic", "anti-inflammatory/neuroprotective", "anti-inflammatory/pro-chondrogenic", "anti-inflammatory/TPP1 enhancing", "anti-inflammatory/TPP1 enhancing/vitamin combination", "pharmacological chaperone", "cytochrome P450 rescue", "enzyme replacement therapy", "enzyme replacement therapy–pharmacological chaperone co-administration", "lentiviral vector", "membrane stabilization", "lysosomal cholesterol redistributor", "replacement therapy with a modified enzyme", "nonviral vector directing transgene integration", "receptor amplification", "retroviral vector", "small molecule facilitating intracellular substrate transport", "stem cells", "stop-codon read-through", and "substrate reduction".

Time to FDA approval was defined as the time period from orphan drug designation until approval by the FDA [1]. Drug approval rates were defined as the proportion of orphan drug designations *approved* out of overall orphan drug designations *granted*. Missing data were not imputed.

## Statistical analysis

Standard techniques of descriptive statistics were applied: continuous variables were summarized with mean, standard deviation, median, minimum and maximum values. Categorical variables were summarized with frequencies and percentages.

Comparative statistics were performed with the appropriate parametric test for data with Gaussian distribution. Differences in mean times to approval (defined as time between orphan drug designation and FDA approval) between drug compound subtypes were analyzed with ANOVA. Differences between frequency counts for approval rates of lysosomal orphan drugs versus approval rates for non-lysosomal orphan drugs were compared with the chi-square test. A two-sided p-value < 0.05 was considered statistically significant.

### The following groups were analyzed:

1. Orphan drug designations and approvals by the FDA (overall and for compounds intended to treat lysosomal storage diseases) by year

2. Orphan drug designations by the FDA for compounds intended to treat lysosomal storage diseases by year and

   a. by disease

   b. by pharmacological technology platform (as specified in the definitions section)

3. Withdrawn FDA orphan drug designations for compounds intended to treat lysosomal storage diseases

4. FDA approved therapies for lysosomal storage disorders

   a. by disease with time periods from orphan drug designation until approval and market exclusivity

   b. by year by pharmacological technology platform (as specified in the definitions section)

All statistical analyses were performed using SAS Enterprise guide 7.13 HF4, SAS Institute Inc., Cary, NC, USA. Graphs were generated with R [16] and GraphPad Prism 5.04, GraphPad Software, Inc., San Diego, CA, USA.

## Results

### The drug development pipeline: Orphan drug designations granted by the FDA

Between 1 January 1983 and 10 May 2019, 124 orphan drug designations were granted by the FDA for compounds intended to treat 28 lysosomal storage diseases (Fig 1A). For comparison, in the same time period, the FDA granted 4979 orphan drug designations overall, out of which 783 were approved (Fig 1B). Twenty lysosomal conditions had multiple orphan drug designations. Most orphan drug designations were granted for Gaucher disease (N = 16), Pompe disease (N = 16), Fabry disease (N = 10), MPS II (N = 10), MPS I (N = 9), and MPS IIIA (N = 9), followed by 14 other diseases depicted in Fig 2A. Eight lysosomal conditions had one orphan drug designation each. Enzyme replacement therapies, gene therapies, small molecules, and other technology platform classes were designated as orphan drugs intended to treat lysosomal storage diseases (Fig 2B). Nine granted orphan drug designations were subsequently withdrawn (Table 1). The reason for withdrawal is not specified in the FDA orphan drug database. The approval rates of *lysosomal* orphan drugs (18.5%) did not differ from approval rates for *non-lysosomal* orphan drugs (15.7%, p = 0.38, chi-square)

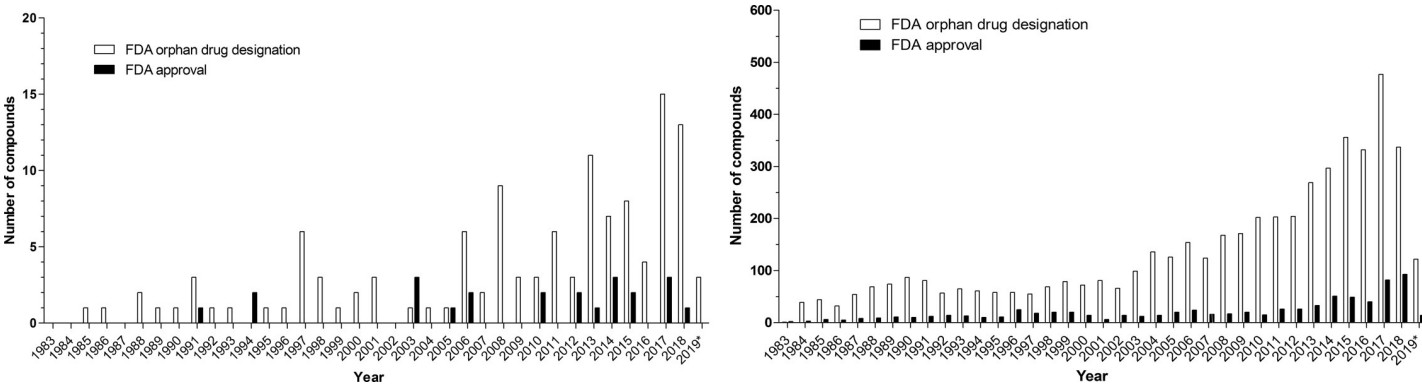

**Fig 1. A**: Number of orphan drug designations (open bars) and FDA approvals (full bars) for compounds intended to treat lysosomal storage diseases by year. * indicates close of database: 10 May 2019. **B**: Overall number of orphan drug designations (open bars) and FDA approvals (full bars) by year. * indicates close of database: 10 May 2019.

## Lysosomal storage disorders with FDA approved therapies

Twenty-three orphan drugs were approved for the treatment of 11 lysosomal storage diseases. Four diseases had multiple therapeutics approved, i.e. Gaucher disease (N = 6), cystinosis (N = 5), Pompe disease (N = 3), and Fabry disease (N = 2), (Fig 3A). The remaining seven diseases had one compound each approved by the FDA (i.e., CLN2, LAL-D, MPS I, II, IVA, VI, VII). CLN2 was the only neuronopathic lysosomal storage disease with an FDA approved therapy directly targeting the brain; all the other therapies address systemic non-neurological manifestations. FDA approved therapies included enzyme replacement therapies (N = 15) and small molecules (N = 8), but no other class of drugs (Fig 3B, Table 2). Approved treatments for

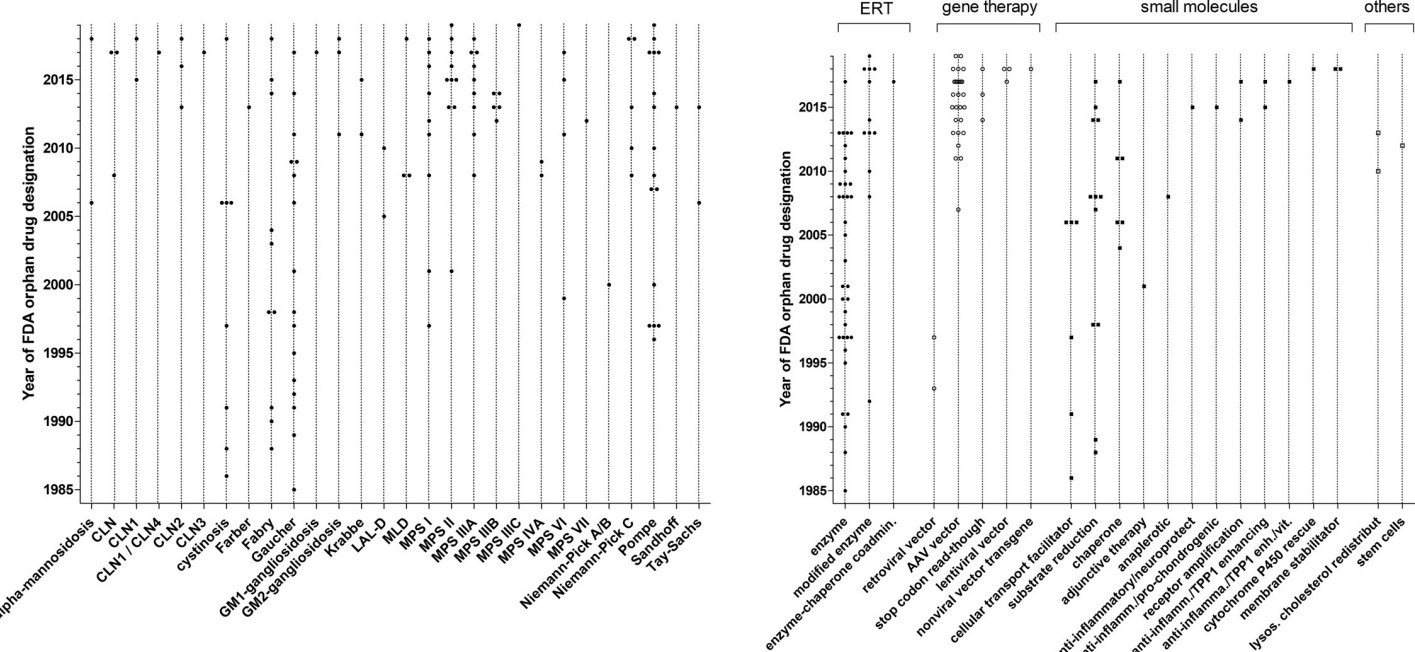

**Fig 2. A**: Orphan drug designations granted by the FDA for compounds intended to treat lysosomal storage disorders by year and specific disease. **B**: Orphan drug designations granted by the FDA for compounds intended to treat lysosomal storage disorders by year and pharmacological technology platform.

**Table 1. Withdrawn orphan drug designations.** Reasons for and time of withdrawal were not specified in the FDA database.

| Compound | Pharmacological subtype | Year of orphan drug designation | Indication under development |
|---|---|---|---|
| Ataluren | Stop-codon read-through | 2014 | Treatment of mucopolysaccharidosis type I |
| Recombinant human alpha-N-acetylglucosaminidase | Enzyme | 2013 | Treatment of mucopolysaccharidosis IIIB (Sanfilippo B syndrome) |
| Recombinant human arylsulphatase A | Enzyme | 2008 | Treatment of metachromatic leukodystrophy (MLD) |
| Miglustat | Substrate reduction | 2008 | Treatment of the neurological manifestations of Niemann-Pick disease, type C |
| Duvoglustat hydrochloride | Substrate reduction | 2007 | Treatment of Pompe disease |
| Isofagomine tartrate | Chaperone | 2006 | Treatment of Gaucher disease |
| Retroviral vector, R-GC and GC gene 1750 | Retroviral vector | 1997 | Treatment of Gaucher disease |
| Human acid precursor alpha-glucosidase, recombinant | Enzyme | 1996 | Treatment of glycogen storage disease type II |
| Phosphocysteamine | Substrate reduction | 1988 | Treatment of cystinosis. |

cystinosis included three different formulations and three different age groups. Enzyme replacement therapies with alglucosidase alfa for Pompe disease were manufactured in two different bioreactor systems and approved for pediatric and adult age groups.

## Regulatory drug development timelines

Overall mean time to approval, defined as time between orphan drug designation and FDA approval was 89.7 SD 55.00 (range 8–203, N = 23) months. Stratified by drug compound

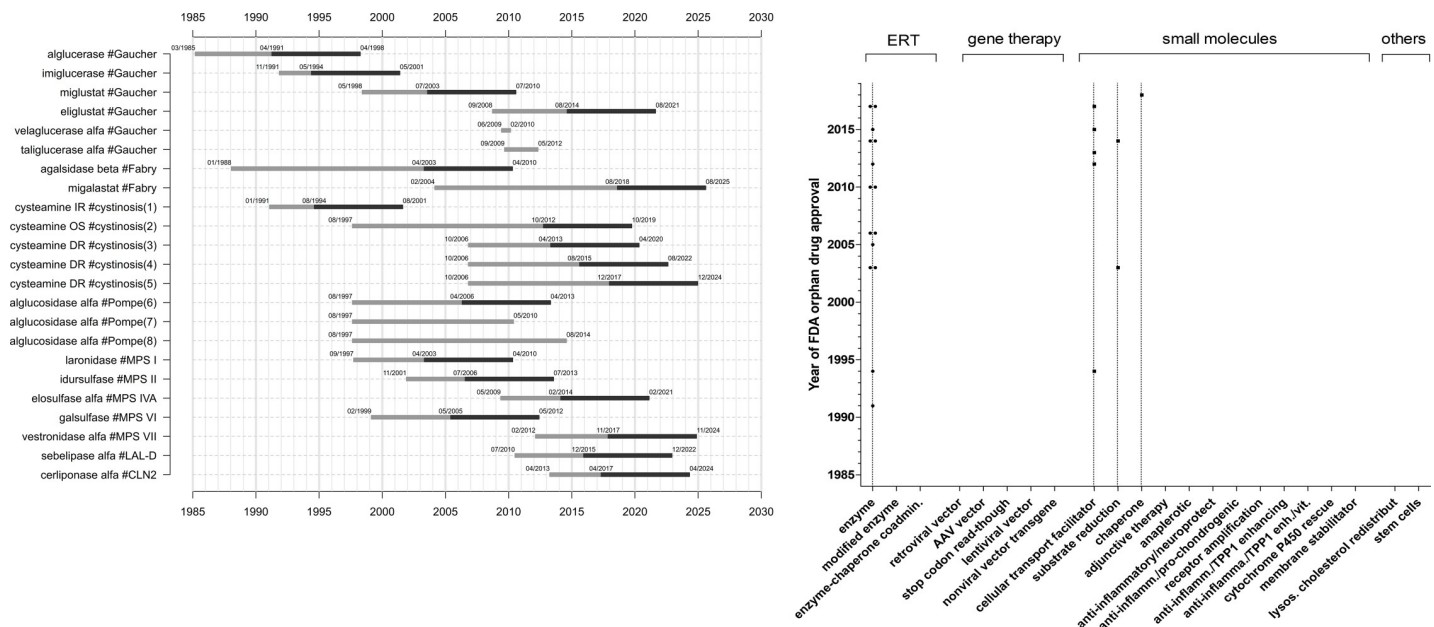

**Fig 3. FDA approved compounds for the treatment of lysosomal storage disorders (depicted as compound #disease), development times and market exclusivity.**
**A**: Grey bars indicate drug development times, i.e. time from orphan drug designation to orphan drug approval by the FDA. Black bars indicate, if applicable, market exclusivity periods. (1)–systemic administration, immediate release (IR). (2)—ophthalmic solution (OS). (3)–systemic administration, delayed release (DR), adults. (4)–systemic administration, delayed release (DR), age 2 to 6 years. (5)—systemic administration, delayed release (DR), age 1 to less than 2 years. (6)—bioreactor 160 L. (7)—bioreactor 4000 L, 8 years and older.(8)—bioreactor 4000 L, all ages. **B**: FDA approved therapies for the treatment of lysosomal storage disorders by year of approval and pharmacological technology platform.

**Table 2. Mechanism of action of FDA approved small molecules (\*) and small molecules in development, intended to treat a lysosomal storage disorder.**

| Mechanism of action | Compound | Disease with FDA orphan drug designation |
|---|---|---|
| **Targeting the affected gene**: Stop-codon read through in missense mutations [36] | | |
| | 6'-(R)-methyl-5-O-(5-amino-5,6-dideoxy-alpha-L-talofuranosyl)-paromamine sulfate | MPS I Cystinosis |
| **Targeting the affected enzyme**: TPP1 enhancer, chaperone, enzyme/chaperone co-administration [37–42] | | |
| | Gemfibrozil | CLN |
| | N-t-butylhydroxylamine | CLN1 |
| | Modified cholera toxin¶ | Gaucher disease |
| | Pyrimethamine | $G_{M2}$-gangiosidosis (Tay-Sachs and Sandhoff disease) |
| | Ambroxol | Gaucher disease |
| | N-acetyl-glucosamine thiazoline | Adult Tay-Sachs disease |
| | Migalastat hydrochloride* | Fabry disease |
| | Recombinant human acid a-glucosidase/miglustat | Pompe disease |
| **Targeting storage**: Substrate reduction and subcellular storage redistribution [43–45] | | |
| | Odiparcil | MPS VI |
| | Lucerastat | Fabry disease |
| | Venglustat | Fabry disease |
| | (3S)-1-azabicyclo[2.2.2]oct-3-yl {2-[2-(4-fluorophenyl)-1,3-thiazol-4-yl]propan-2-yl} carbamate | Gaucher disease |
| | 2-hydroxypropyl-B-cyclodextrin§ | Niemann-Pick disease type C |
| | Hydroxy-Propyl-Beta-Cyclodextrin§ | Niemann-Pick disease type C |
| | Miglustat* | Gaucher disease* |
| | Eliglustat* | Gaucher disease type I* |
| | Cysteamine* | Cystinosis* NCL (Batten disease) |
| | 1,5-(Butylimino)-1,5 dideoxy,D-glucitol | Fabry disease |
| | L-cycloserine | Gaucher disease |
| **Targeting cellular uptake of therapeutic enzymes**: Receptor amplification [46] | | |
| | Clenbuterol | Pompe disease |
| **Mitigation of cellular damage** (Antiinflammatory, pro chondrogenic, neuroprotective cytochrome P450 rescue) or **anaplerotic** [20, 37, 47–49] | | |
| | Ursodeoxycholic acid | Niemann-Pick C |
| | Gemfibrozil and vitamin A | CLN |
| | Ibudilast | Krabbe disease |
| | Pentosan polysulfate sodium | MPS VI |
| | Triheptanoin | Pompe disease |

¶protein acting as a chaperone

§polymer

subtypes, mean time to approval for enzyme replacement therapies was 81.2 SD 56.42 (range 8–203, N = 15) months, mean time to approval for small molecules facilitating subcellular transport was 107.8 SD 52,96 (range 40–181, N = 5) months, and mean time to approval for substrate reduction therapies was 66.5 SD 6.36 (range 62 to 71, N = 2) months. Time to

approval for the pharmacological chaperone therapy was 173 months. Differences between the above groups were not statistically significant (p = 0.33, ANOVA). The drug development timelines and market exclusivity periods, an incentive granted by the FDA to stimulate orphan drug development [13], are illustrated in Fig 3A.

## Discussion

By 10 May 2019, 23 orphan drugs were approved by the FDA for the treatment of 11 lysosomal storage disorders. This is an increase of nine more approved orphan drugs and four more treatable lysosomal disease (i.e. CLN2, MPS VII, LAL-D, and MPS IVA) compared to 2013 [1].

While alglucerase for Gaucher disease was the first orphan drug approved for a lysosomal storage disease in 1991, intrathecally administered cerliponase alfa for CLN2, FDA approved in 2017, is the first orphan drug for a lysosomal storage disorder to directly treat the brain, which is a significant therapeutic innovation [17, 18]. Since 2013, 54 more orphan drug designations were granted. In addition, diseases such as CLN1, CLN3, CLN4, Farber disease, and $G_{M1}$-gangliosidosis did not have orphan drug designations in 2013. This indicates that drug development in lysosomal storage disorders is now being driven into mainly neuronopathic conditions (Fig 2A). The overall growth curve of orphan drug designations appears to accelerate over time and may become exponential (Fig 1A), possibly following a global trend (Fig 1B). Interestingly, the drug approval rate in lysosomal orphan drug development and non-lysosomal orphan drug development did not differ. Technology is evolving: while enzyme replacement therapies had initially set the trend, more modified enzymes, including fusion proteins, and an enzyme-chaperone co-administration entered the development pipeline. This may be a reaction to the increasing recognition in the field that, in general, systemically administered enzyme replacement therapy with conventional enzymes can easily access organs such as liver and spleen but have little impact on bone and CNS manifestations. Four small molecules have been approved by the FDA for the treatment of a lysosomal storage disease. Their mechanisms of action target the facilitation of subcellular transport (e.g., cysteamine for cystinosis, approved in 1994) and the reduction of storage (miglustat, approved in 2003, and eligustat, approved in 2014, both for the treatment of Gaucher disease) [1]. In 2018, migalastat, which stabilizes the misfolded enzyme alpha-galactosidase A, was approved as a first-of-its kind pharmacological chaperone by the FDA for the treatment of Fabry disease [19], (Table 2). Mechanisms of action for small molecules, either approved or in drug development, encompassed the broad spectrum of underlying pathophysiology and aimed at 1) targeting the affected gene, 2) targeting the affected enzyme, 3) targeting storage, 4) targeting cellular uptake of therapeutic enzymes, and 4) mitigating the cellular damage or anaplerotic (Table 2). It is possible and likely that not all mechanistically meaningful approaches lead to clinical benefit in patients [20]. The plethora of innovative ideas for pharmacological approaches is laudable but should not lead a treating physician to engage in off-label use but rather encourage international collaboration aimed to generate the highest standard of evidence-based knowledge by respecting excellence clinical research [21].

Gene therapy now plays a larger role in the drug development pipeline compared to the situation in our last analysis [22]. This may again be a reaction to the increased recognition of enzyme replacement therapies' substantial limitations, as described in detail above. The technical approach towards gene therapy is evolving as illustrated in Fig 2B. Gene therapies rely, at least in principle, on the assumption that a single treatment may result in a sustained, potentially curative clinical benefit for the patients. The first molecular tools enabling efficient nontoxic gene transfer into human somatic cells were recombinant replication-deficient vectors [23]. Among those, retroviral and adeno-associated viral (AAV) vectors have been the most

widely used in particular for *ex vivo* T cell engineering or genetically engineered hematopoietic stem cells (HSCs) for the treatment of primarily hematologic or oncologic conditions such as pediatric ALL, β-thalassemia or adenosine deaminase deficiency [24–26]. In contrast, while the first two orphan drug designations for gene therapy for lysosomal storage diseases in 1993 and 1997 (both for Gaucher disease) relied on retroviral vectors, this platform was subsequently abandoned. This is likely due to the emergence of serious toxicities related to high gene transfer including insertional genotoxicity, immune destruction of genetically modified cells, and immune reactions related to the application of certain vectors [27, 28]. The next step in gene technology was the introduction of AAV (designated for Pompe disease in 2007), followed by stop-codon read-through (designated in 2014 and 2016 for MPS I, and in 2018 for cystinosis). Moreover, lentiviral vector (designated in 2018 for Fabry disease and MLD), and nonviral vector directing transgene integration (designated in 2018 for MPS I) technologies are being considered, all of which have to prove their safety and efficacy in the future. More sophisticated genome editing technologies that enable a variety of therapeutic genome modifications (gene addition, gene ablation or "gene correction") consist of the administration of transcription activator-like effector nucleases (TALENs) and or CRISPR-Cas 9 system to efficiently cleave and modify DNA at sites of interest [29–33]. Those approaches are currently limited to applications in basic research, but transfer into clinical trials can be expected in the near future [34, 35]. Until close of database no gene therapy was approved for the treatment of lysosomal storage disorders (Fig 3B). If proven successful in registration trials—which would in all likelihood be small clinical trials of a limited duration—it is of particular interest, how long the therapeutic effect of gene therapy can be sustained during a patients' lifetime, and, if this time is limited, whether it would be safe and feasible to repeat the administration of a gene therapy multiple times in a single patient. It is anticipated that novel therapies will be costly. Important topics for future investigations will include patient selection, starting and stopping criteria.

For the correct contextual interpretation it is important to be aware of important limitations of the present work as pointed out previously [1]. Orphan drug designation was considered as the expressed intent to develop a drug. This may be influenced by strategic and patent related considerations. Not all manufacturers may choose such a publicly visible pathway at an early stage. Therefore, time to approval as presented here may be biased by the intellectual property strategy of the respective drug development program. Orphan drug development outputs in jurisdictions other than the FDA were, as in our previous analysis, not taken into account because this analysis was by definition focused on the impact of the US orphan drug act [1, 13]. As drug development in lysosomal storage disorders is, in general, a global enterprise we consider the present findings generalizable within the context of their limitations.

## Conclusions

Activities in orphan drug development for lysosomal storage disorders are steadily increasing, which follows a global trend in orphan drug development overall. Newly approved products included "me-too"–enzymes, and also innovative compounds such as the first ERT targeting the brain in CLN2 and the first-of-its-kind pharmacological chaperone for the treatment of Fabry disease. The drug development pipeline for LSDs is growing and evolving, with increased focus on diverse small-molecule targets and gene therapy.

## Supporting information

**S1 Checklist. STROBE statement.**
(DOC)

## Acknowledgments

We thank Lorna Stimson, PhD, for language editing.

## Author Contributions

**Conceptualization:** Markus Ries.

**Data curation:** Sven F. Garbade.

**Formal analysis:** Sven F. Garbade, Matthias Zielonka, Konstantin Mechler, Stefan Kölker, Georg F. Hoffmann, Christian Staufner, Eugen Mengel, Markus Ries.

**Investigation:** Markus Ries.

**Methodology:** Markus Ries.

**Software:** Sven F. Garbade.

**Supervision:** Markus Ries.

**Validation:** Markus Ries.

**Writing – original draft:** Markus Ries.

**Writing – review & editing:** Sven F. Garbade, Matthias Zielonka, Konstantin Mechler, Stefan Kölker, Georg F. Hoffmann, Christian Staufner, Eugen Mengel.

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
