## [Decision Letter · Decision Letter 0]

25 Feb 2020

PONE-D-20-00658

FDA orphan drug designations for lysosomal storage disorders - a cross sectional analysis

PLOS ONE

Dear Dr Ries,

Thank you for submitting your manuscript to PLOS ONE. After careful consideration, we feel that it has merit but does not fully meet PLOS ONE’s publication criteria as it currently stands. Therefore, we invite you to submit a revised version of the manuscript that addresses the points raised during the review process.

Although the two reviewers have a favorable view of the scientific content, they both, and in particular Reviewer 2, are very unhappy with the format of the manuscript, and found it difficult to read. So the revised version should be reformatted so as to address their specific concerns.

We would appreciate receiving your revised manuscript by Apr 10 2020 11:59PM. To enhance the reproducibility of your results, we recommend that if applicable you deposit your laboratory protocols in protocols.io, where a protocol can be assigned its own identifier (DOI) such that it can be cited independently in the future. For instructions see: http://journals.plos.org/plosone/s/submission-guidelines#loc-laboratory-protocols

We look forward to receiving your revised manuscript.

Kind regards,

Israel Silman

Academic Editor

PLOS ONE

Journal Requirements:

"The authors received no specific funding for this work."

We note that one or more of the authors are employed by a commercial company: SphinCS GmbH.

Reviewers' comments:

Reviewer's Responses to Questions

**Comments to the Author**

1. Is the manuscript technically sound, and do the data support the conclusions?

Reviewer #1: Yes

Reviewer #2: Partly

2. Has the statistical analysis been performed appropriately and rigorously? 

Reviewer #1: Yes

Reviewer #2: I Don't Know

3. Have the authors made all data underlying the findings in their manuscript fully available?

Reviewer #1: Yes

Reviewer #2: Yes

4. Is the manuscript presented in an intelligible fashion and written in standard English?

Reviewer #1: Yes

Reviewer #2: No

5. Review Comments to the Author

Reviewer #1: The review explains the state of the art of development of treatments for LSDs. It is well written and with complete information.

The manuscript is well organized and clearly written.

I would suggest to change the figures, the points alone in them are difficult to understand. May be if you put a line, it could be better understandable.

Reviewer #2: The manuscript by Garbade et al, entitled "FDA orphan drug designations for lysosomal storage disorders - a cross sectional analysis" represents a useful summary and analysis of all orphan drug designation related to lysosomal storage disorders, compared to all orphan drug designations in the same period of time.

Although well-written, the manuscript presents a strange form, since Authors, instead of describing their results citing related Figures and Tables, as commonly performed, here present their results mainly using the Figure and Table legends, which renders results very difficult to read. Therefore, presentation of results should be totally re-written. In addition, Authors should provide all Figure and Table legends in a section separated from the main text. In its present form, the paper is really hard to understand and quite confusing.

The statistical analysis performed, although described in the methods section of the manuscript, is not clearly described in relation to the specific groups of data analyzed.

The name of the last Figure reported is Figure 1B instead of Figure 3B, please correct.

In addition, few English mistakes are present in the manuscript, please check and correct.

6. PLOS authors have the option to publish the peer review history of their article (what does this mean?). If published, this will include your full peer review and any attached files.

Reviewer #1: No

Reviewer #2: No

---

## [Author Response · Author response to Decision Letter 0]

9 Mar 2020

please refer to the uploaded document "response to reviewers"

---

## [Editor Report · Decision Letter 1]

12 Mar 2020

FDA orphan drug designations for lysosomal storage disorders – a cross-sectional analysis

PONE-D-20-00658R1

Dear Dr. Ries,

We are pleased to inform you that your manuscript has been judged scientifically suitable for publication and will be formally accepted for publication once it complies with all outstanding technical requirements.

With kind regards,

Israel Silman

Academic Editor

PLOS ONE
---

## [Editor Report · Acceptance letter]

18 Mar 2020

PONE-D-20-00658R1 

FDA orphan drug designations for lysosomal storage disorders – a cross-sectional analysis 

Dear Dr. Ries:

I am pleased to inform you that your manuscript has been deemed suitable for publication in PLOS ONE. Congratulations! Your manuscript is now with our production department. 

With kind regards,

on behalf of

Prof. Israel Silman 

Academic Editor

PLOS ONE